# Effects and Safety of Wearable Exoskeleton for Robot-Assisted Gait Training: A Retrospective Preliminary Study

**DOI:** 10.3390/jpm13040676

**Published:** 2023-04-18

**Authors:** Gwang-Min Park, Su-Hyun Cho, Jun-Taek Hong, Dae-Hyun Kim, Ji-Cheol Shin

**Affiliations:** 1Department and Research Institute of Rehabilitation Medicine, Yonsei University College of Medicine, Seoul 03722, Republic of Korea; 2Department of Physical and Rehabilitation Medicine, Center for Prevention and Rehabilitation, Heart Vascular Stroke Institute, Samsung Medical Center, Seoul 06355, Republic of Korea

**Keywords:** robot-assisted gait training, wearable robot, rehabilitation, gait

## Abstract

Background: Wearable devices for robot-assisted gait training (RAGT) provide overground gait training for the rehabilitation of neurological injuries. We aimed to evaluate the effectiveness and safety of RAGT in patients with a neurologic deficit. Methods: Twenty-eight patients receiving more than ten sessions of overground RAGT using a joint-torque-assisting wearable exoskeletal robot were retrospectively analyzed in this study. Nineteen patients with brain injury, seven patients with spinal cord injury and two patients with peripheral nerve injury were included. Clinical outcomes, such as the Medical Research Council scale for muscle strength, Berg balance scale, functional ambulation category, trunk control tests, and Fugl–Meyer motor assessment of the lower extremities, were recorded before and after RAGT. Parameters for RAGT and adverse events were also recorded. Results: The Medical Research Council scale scores for muscle strength (36.6 to 37.8), Berg balance scale (24.9 to 32.2), and functional ambulation category (1.8 to 2.7) significantly improved after overground RAGT (*p* < 0.05). The familiarization process was completed within six sessions of RAGT. Only two mild adverse events were reported. Conclusions: Overground RAGT using wearable devices can improve muscle strength, balance, and gait function. It is safe in patients with neurologic injury.

## 1. Introduction

Gait disorder is a serious long-term disability in adults with neurologic deficit. Gait function is often impaired in people with neurologic deficits, resulting in reduced mobility, which is associated with decreased life satisfaction and a lower quality of life [1]. Additionally, a lack of physical activity increases the risk of developing secondary health problems, such as cardiopulmonary complications, bowel/bladder dysfunction, obesity, osteoporosis, and pressure ulcers [2,3,4,5,6]. These issues can further decrease life expectancy in affected patients [2,3]. Restoration of walking ability is one of the main focuses of rehabilitation in such patients [7,8]. Various rehabilitation strategies, including conventional physical therapy using physical effort, hydrotherapy, and electrical stimulation therapy, have been developed to improve gait ability in these patients [9]. Repetitive task-specific training is a modern concept of rehabilitation to restore walking ability [8,10,11,12]. Previous studies have demonstrated that high intensities of walking training resulted in better outcomes [10,12]. Robot-assisted gait training (RAGT) can provide more supportive high-intensity task-specific training, even in patients with neurologic injury [8]. The benefits of RAGT in patients with neurologic injury are well-established in the literature, and it shows significant improvements in clinical outcomes compared to conventional gait training [13,14,15].

The Cochrane Review for RAGT after stroke was first published in 2007 and has been updated thrice since then [10,16,17,18]. The review in the last update in 2020 showed that RAGT helped more patients walk independently. Zhang et al. showed that RAGT could help patients with spinal cord injury in improving their locomotor ability [19]. In some studies with patients with cerebral palsy, RAGT appears to have a positive effect on locomotor ability and capability for daily activities [20,21]. Other previous studies have also demonstrated the effectiveness of RAGT in various patients [22,23,24]; however, the role of the type of device was unclear [10]. The main robotic design for RAGT consists of an exoskeleton and an end-effector. The exoskeleton resembles human legs, while the robot’s joints usually correspond to the human’s joints and guide the legs by enforcing determined gait postures. The end-effector present in the robot’s footplates is connected to the patients’ feet. It simulates a normal gait by moving the robot’s footplates to the trajectories of the gait cycle [25]. Recently, a wearable type of lightweight exoskeleton for RAGT that can be carried has been developed, enabling patients to walk overground [9]. The wearable device needs energy-efficient and functional purposes [26]. Wearable devices may be more advantageous than static exoskeletons in overground RAGT in restoring walking ability by enabling walking over a hard surface and stairs. Overground RAGT can emulate overground human neuromotor control of locomotion and provide more task-specific training for locomotion compared with a static exoskeleton. Furthermore, overground RAGT using a wearable exoskeletal robot could improve patients’ active balance control, weight transfer, and muscle activation [9]. Balance is an important component of gait function, as it is necessary for maintaining an upright posture and stable walking. Overground RAGT can have a positive effect on balance, as it promotes dynamic postural control, which is critical for an independent gait [27]. Wearable exoskeletons allow for a normal gait in a more outdoor setting, with the patients being able to walk overground and explore the environment [28]. Overground robotic-assisted gait training has the advantage of providing greater freedom of movement during ambulation, opportunities for independent training at home, and the possibility to train more daily living activities, such as sitting, turning, and climbing stairs. Some wearable exoskeletons have already obtained United States Food and Drug Administration approval and/or European Conformity (CE) mark certification and are commercially available [29]. However, limited studies on the effectiveness and safety of the wearable type have been reported for overground RAGT in neurologic deficit [30]. Therefore, we investigated the effectiveness and safety of wearable exoskeletons in patients with neurologic deficit during overground RAGT. Additionally, changes in the robotic parameters during each RAGT session were also analyzed.

## 2. Materials and Methods

### 2.1. Patients and Study Design

This retrospective study reviewed the medical records of adult patients who underwent inpatient rehabilitation and RAGT between 1 November 2020 and 30 April 2022. Twenty-eight patients were selected for the analysis based on the following inclusion criteria: (a) overground RAGT using Angel Legs M performed more than 10 times during the inpatient rehabilitation program; (b) a clinical evaluation performed before and/or after RAGT; and (c) details of the diagnosis and adverse events of RAGT recorded during the inpatient rehabilitation program. Exclusion criteria were: (a) the presence of progressive neurologic disease and (b) coexisting neurological and/or orthopedic disease that could affect gait training. The patients were carefully selected to ensure that they met the criteria for participation in the study and to minimize potential confounding factors.

### 2.2. Wearable RAGT

The Angel Legs M (ANGEL ROBOTICS Co., Ltd., Seoul, Republic of Korea) was used for overground RAGT in the study (weight 19.5 kg). This robot can be used in various individuals with partial impairment of gait function: stroke, spinal-cord injury, neuromuscular diseases, etc. The overground RAGT was conducted for 10–20 sessions per patient, 30 min per session and 5 times a week. All training sessions were conducted under the supervision of a physical therapist. If the patient could not endure 30 min, we stopped the session and recorded the actual performed session time. The robot can provide assistive torque during the different gait phases, which are automatically sensed through combined information from ground contact sensors, encoders (incremental and absolute) in the actuators, and inertial-measurement-unit sensors. According to a polynomial assistive joint torque profile based on the gait phases, flexion torque at the hip and knee joints is generated in the swing phase, and extension torque at the hip and knee joints is generated in the stance phase to help the patient’s gait. More details have been described previously [9]. We collected data on the actual performed session time; cadence in each session (steps/min); and maximal assist power (Nm) of hip flexion, hip extension, knee flexion, and knee extension during each session of the RAGT. Patients walked at a self-selected speed during the treatment sessions, though they were encouraged to keep the speed as fast as possible. The maximal assist power was adjusted to emulate a normal gait pattern according to a therapist’s inspection. We also reviewed medical records for any adverse event reported during the RAGT. We confirmed the completeness of the familiarization process by using the actual performed time of RAGT and the cadence reaching the plateau.

### 2.3. Clinical Evaluation

Clinical evaluations were conducted before and after RAGT. Clinical evaluations included the adjusted Medical Research Council (MRC) scale for muscle strength, Berg balance scale (BBS), functional ambulation category (FAC), trunk control tests (TCT) and Fugl–Meyer motor assessment of the lower extremities (FMLE) [31,32,33,34]. The adjusted MRC scale for muscle strength was the sum of the strength of six lower limb muscle groups, such as hip flexion/extension, knee flexion/extension, and ankle dorsiflexion/plantarflexion, ranging from 0 to 60 [35]. The clinical evaluation before and after RAGT only included an interval of three days.

### 2.4. Statistics

All statistical analyses were performed using R software (version 3.5.1, R Foundation for Statistical Computing, Vienna, Austria). A parametric paired t-test was used for normally distributed measures as tested using the Shapiro–Wilk normality test. For nonparametric data, a Wilcoxon singed-rank test was used. A significance level of *p* < 0.05 was set, indicating that any results with a *p*-value under 0.05 would be considered statistically significant.

## 3. Results

### 3.1. Patient Characteristics

The study consisted of 28 patients who were selected based on their medical history and physical condition. The mean age of the patients was 43.9 ± 22.4 years (range, 19–86 years), and the sample included 17 male and 11 female patients. Of the 28 patients, 19 (68%) patients with brain injury, 7 (25%) with spinal cord injury, and 2 (7%) with peripheral nerve injury were included in this study. Out of 19 patients with brain injury, 7 had stroke, 9 had cerebral palsy, and 3 had others brain injuries, such as encephalitis or tumors. The detailed characteristics of the selected patients are shown in Table 1.

### 3.2. Parameters for Wearable RAGT

The average actual performed session time was 22.9 min in the first session, which reached 30 min within five sessions of RAGT. The average cadence was 33.8 steps/min in the first session; this rose to 46.0 steps/min within six sessions of RAGT. However, individual variations were seen until the 20th session (Figure 1). The averages of the maximal assist power were 7.7 Nm in hip flexion, 9.2 Nm in hip extension, 7.7 Nm in knee flexion, and 9.2 Nm in knee extension. A decreasing trend of the maximal assist power for hip flexion was observed throughout the sessions (Figure 2).

### 3.3. Changes in Clinical Outcome after RAGT

The results of the study showed that after overground RAGT, there was a significant improvement in MRC (36.6 ± 2.1 to 37.8 ± 2.4, *p* = 0.012), BBS (24.9 ± 3.3 to 32.2 ± 3.2, *p* = 0.001), and FAC (1.8 ± 0.4 to 2.7 ± 0.3, *p* = 0.030) scores in all patients. In patients with brain injury, there was a significant improvement in MRC (39.4 ± 1.5 to 40.8 ± 1.5, *p* = 0.017) and BBS (23.5 ± 3.6 to 30.6 ± 3.2, *p* = 0.001) scores after overground RAGT. However, no significant improvement was seen in any of the clinical outcomes in SCI patients after RAGT. The changes in clinical outcomes are shown in Table 2.

### 3.4. Feasibility of Wearable RAGT

Only two patients felt pain during the first session. Their skin was tender from the straps used to attach their leg to the exoskeletal robot’s leg. Based on the numeric pain intensity scale, they scored 1 and 3 points, respectively. The pain was reduced by adjusting the position of the strap. Patients did not take any pain-relief medication due to the diminishing intensity of pain after the cessation of the treatment session. None of the patients stopped their session due to pain during RAGT.

## 4. Discussion

Robot-assisted gait training (RAGT) emerged nearly 30 years ago. Since the development of the Locomat in 1994, the technology for robot devices used in gait training has advanced rapidly [36]. RAGT provides supportive high-intensity task-specific training for patients with neurologic injury and can improve a patient’s gait and balance function. Robot-assisted gait training (RAGT) provides supportive high-intensity task-specific training for patients with neurologic injury. This study evaluated the effectiveness and safety of a wearable robotic exoskeleton in patients with neurologic injuries during overground RAGT. Significant improvements in clinical outcomes, assessed using the MRC, BBS, and FAC scales, were observed after overground RAGT. The actual performance time of RAGT and the cadence reached six sessions, and the maximal torque assistance for hip flexion showed a decreasing trend throughout the sessions. The adverse events were minor, and no patient stopped RAGT due to adverse events.

In our study, overground RAGT improved clinical outcomes associated with gait function. These results are in line with those of previous studies of static RAGT on treadmills that demonstrated an improved walking ability [8,37]. The advantage of overground RAGT is that it is more effective in dynamic postural control than static RAGT [9,38]. Overground RAGT facilitates the trunk muscles by shifting the body weight according to the gait movement without a body-weight support system, which is mainly used in static RAGT on a treadmill [38]. Dynamic postural control, which is affected by trunk muscles, is critical for an independent gait; hence, in this study, overground RAGT showed an improved effectiveness in terms of BBS and FAC measures. Furthermore, the joint-torque-assisting system can promote patients’ motivation and improve muscle strength by assisting patients in their effort to move their joints [9]. Therefore, muscle strength, reflected by MRC, improved after overground RAGT. Since during the swing phase, the hip flexor needs more muscle power to lift the limb with the exoskeleton, the Angel Legs M system may facilitate the muscle strength of the hip flexor. The assist torque for hip flexion decreased throughout sessions due to the increasing hip flexor muscle strength.

For the wearable RAGT, all patients completed the familiarization process within several sessions. Due to the absence of weight support systems, the wearable RAGT can be challenging for patients during gait training. The safety of the wearable RAGT was clearly seen in this study, since no severe adverse events were reported; only mild pain was felt by two patients, but it did not interrupt their session. Previous studies on similar wearable devices have demonstrated the safety of gait training [38,39]. However, when using wearable RAGT, the absence of a weight support system demands precise clinical indications for patients who can control their trunk posture. Furthermore, at least one supervisor is required to closely monitor the use of overground RAGT, especially during the familiarization process. In contrast to conventional physical therapy, where continuous physical assistance from a therapist is needed, a wearable RAGT is more efficient because the therapist performs a supervisory role.

There are a few limitations in this study. The small heterogenous sample size, retrospective design of the study, and absence of a control group are potential confounding factors. Our study should be interpreted with caution because it has a small sample size and is not a randomized controlled trial. Despite these limitations, to the best of our knowledge, this is the first study to evaluate the effectiveness, safety, improvements in clinical outcomes, and changes of robotic parameters following overground RAGT in patients with neurologic deficits using a wearable robotic exoskeleton. Until now, research on RAGT for peripheral nerve injury patients suffering from conditions such as chronic inflammatory demyelinating polyradiculoneuropathy (CIDP) or Guillain-Barre Syndrome (GBS), in addition to stroke or SCI, has not been widely conducted. This study is the first to analyze the effectiveness of overground RAGT. In the future, it will be necessary to recruit more patients with peripheral nerve injuries to validate these results. Thus, prospective randomized controlled studies with larger numbers of patients are needed to confirm the effectiveness and safety of RAGT in patients with various clinical conditions.

## 5. Conclusions

This study showed that overground RAGT using a wearable type is feasible in patients with neurologic injury, enabling them to reach a tolerable level within six sessions of RAGT. Moreover, the wearable joint-torque-assisting robot for RAGT can improve muscle strength, balance, and gait function by providing overground high-intensity gait-specific training. Furthermore, the application of a wearable type for RAGT is safe in patients with neurologic deficits. Therefore, our study can be used in future studies when setting up well-defined protocols to provide the best patient-specific rehabilitation training, especially for patients with brain injuries.

## Figures and Tables

**Figure 1 jpm-13-00676-f001:**
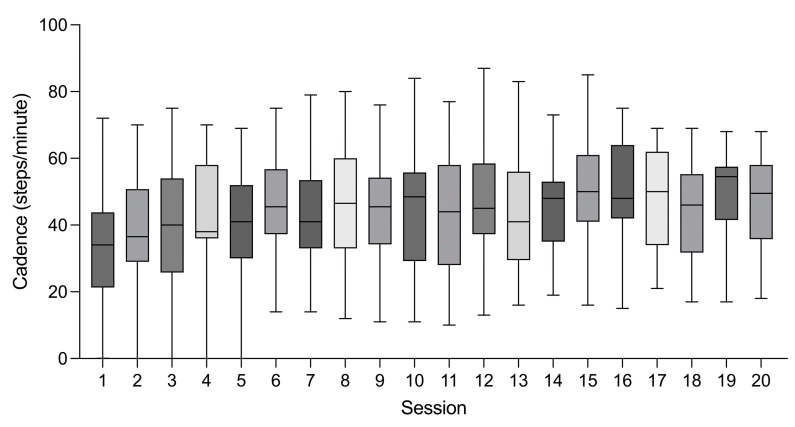
Cadence in each session of overground robot-assisted gait training. Boxes denote mean ± standard deviation, and the vertical bars represent the range.

**Figure 2 jpm-13-00676-f002:**
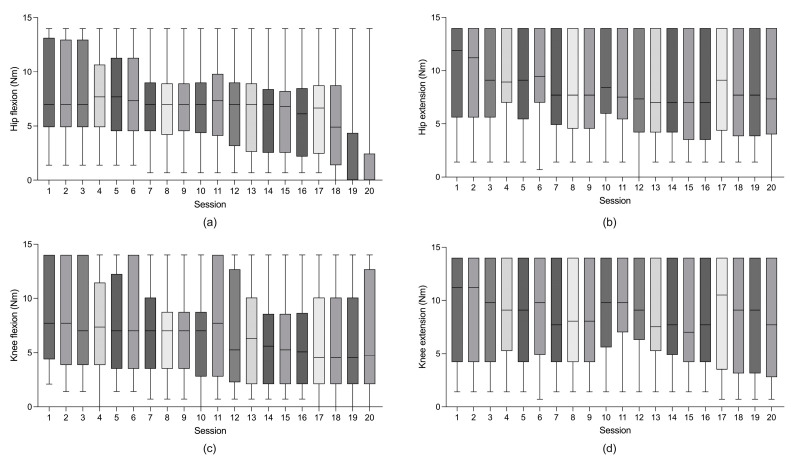
Maximal joint-assisting torque in each session of overground robot-assisted gait training. (**a**) Maximal hip flexion torque (Nm); (**b**) maximal hip extension torque (Nm); (**c**) maximal knee flexion torque (Nm); and (**d**) maximal knee extension torque (Nm). Boxes denote mean ± standard deviation, and the vertical bars represent the range.

**Table 1 jpm-13-00676-t001:** General characteristics of included patients.

Variables	Values
Demographics	
Mean age (years, mean ± SD)	43.9 ± 22.4
Male:Female (*n*, %)	17 (60.7):11 (39.3)
Height (cm, mean ± SD)	167.1 ± 10.5
Weight (kg, mean ± SD)	64.4 ± 12.4
Diagnosis (*n*, percentage)	
Brain injury	19 (68.0)
Stroke	7 (25.0)
Cerebral palsy	9 (32.1)
etc.	3 (10.7)
Spinal cord injury	7 (25.0)
Trauma	2 (7.1)
Tumor	3 (10.7)
etc.	2 (7.1)
Peripheral nerve injury	2 (7.1)
CIDP	2 (7.1)

**Table 2 jpm-13-00676-t002:** Changes in clinical outcomes before and after RAGT.

Clinical Measure	Values		*p*-Value
	Before RAGT	After RAGT	
All patients			
MRC *	36.6 ± 2.1	37.8 ± 2.4	0.012 ^†^
BBS **	24.9 ± 3.3	32.2 ± 3.2	0.001 ^†^
FAC *	1.8 ± 0.4	2.7 ± 0.3	0.030
TCT	59.4 ± 4.0	79.4 ± 5.8	0.057
Brain injury			
MRC *	39.4 ± 1.5	40.8 ± 1.5	0.017 ^†^
BBS **	23.5 ± 3.6	30.6 ± 3.2	0.001 ^†^
FAC	1.8 ± 0.5	2.6 ± 0.4	0.053
TCT	59.4 ± 4.0	79.4 ± 5.8	0.057
FMLL	13.0 ± 3.8	22.6 ± 3.4	0.062
Spinal cord injury			
MRC	22.2 ± 6.6	24.6 ± 8.3	0.586
BBS	25.3 ± 7.5	34.0 ± 11.2	0.371

Values shown as average ± standard error. * *p* < 0.05, ** *p* < 0.01; MRC, adjusted Medial Research Council scale for muscle strength; BBS, Berg balance scale; FAC, functional ambulation category; TCT, trunk control test. ^†^ *p* value by paired t-test.

## Data Availability

The data presented in this study are available on request from the corresponding author. The data is not publicly available due to privacy.

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
