# Peer review of "Effects and Safety of Wearable Exoskeleton for Robot-Assisted Gait Training: A Retrospective Preliminary Study"

_jpm, 2023, doi:10.3390/jpm13040676_

Round 1
Reviewer 1 Report
This is a well-written and informative study on the effectiveness and safety of wearable exoskeletons for robot-assisted gait training in patients with neurologic deficits. However, there are a few areas where the authors could improve the manuscript:
1.The introduction provides a clear and concise background to the study, however, it could be improved by providing more information on the current state of wearable exoskeletons and their role in gait rehabilitation.
2.The methods section is well written and provides a detailed description of the study design and participant selection criteria. However, it could be improved by providing more information on the specific type of wearable exoskeleton used and the training protocols. In addition, there is a lack of detailed explanation and display of the wearable exoskeleton used in the experiment.
3.The discussion section of the manuscript could be expanded to provide a more detailed interpretation of the findings and their implications for clinical practice. The authors could also discuss areas for future research and how their findings could be used to guide the development of new wearable exoskeletons for gait training.
4.The authors mention that parameters for RAGT were recorded, but they do not provide any information on what these parameters were or how they were used to guide the training. This information would be valuable for researcher who may be interested in replicating the study.
5.The author reported that only two mild adverse events were reported and explained in the manuscript that the event caused mild pain in the patient during the experiment, but there were no details on how to handle it.This information is important for understanding the safety of wearable exoskeletons and should be included in the manuscript.
6.The sample size for this study was relatively small, including only 29 patients. The authors acknowledge this limitation, but lack a discussion on how it affects the extensibility of their findings.
Overall, this is a valuable contribution to the literature on wearable exoskeletons for gait training, but the authors could improve the manuscript by addressing the above concerns.
Author Response
Response to Reviewer 1 Comments
Point 1: The introduction provides a clear and concise background to the study, however, it could be improved by providing more information on the current state of wearable exoskeletons and their role in gait rehabilitation.
Response 1: Thank you for the comment. We have included some references to inform the current state of wearable exoskeletons and their role in gait rehabilitation as below.
Line 53-60: Wearable exoskeletons allow normal gait in a more outdoor setting, with the patients able to walk overground and to explore the environment[15]. Overground robotic-assisted gait training has the advantage of providing greater freedom of movement during ambulation, opportunities for independent training at home, and the possibility to train more activities of daily living such as sitting, turning, and climbing stairs. Some wearable exoskeletons have already obtained United States Food and Drug Administration approval and/or European Conformity (CE) mark certification and are commercially available[16].
Point 2: The methods section is well written and provides a detailed description of the study design and participant selection criteria. However, it could be improved by providing more information on the specific type of wearable exoskeleton used and the training protocols. In addition, there is a lack of detailed explanation and display of the wearable exoskeleton used in the experiment.
Response 2: Thank you for the comment. We have added detailed explanation of wearable exoskeleton to help understanding as below.
Line 81-87: The robot can provide assistive torque during the different gait phases, which are automatically sensed through combined information from ground contact sensors, encoders (incremental and absolute) in the actuators, and inertial measurement unit sensors. According to a polynomial assistive joint torque profile based on the gait phases, flexion torque at the hip and knee joints is generated in the swing phase, and extension torque at the hip and knee joints is generated in the stance phase to help the patient’s gait. More details have been described previously[13].
Point 3: The discussion section of the manuscript could be expanded to provide a more detailed interpretation of the findings and their implications for clinical practice. The authors could also discuss areas for future research and how their findings could be used to guide the development of new wearable exoskeletons for gait training.
Response 3: Thank you for the comment. We have added the usability of our study as below.
Line 204-206: Therefore, it can be used when establishing well-defined protocols to provide the best patient-specific rehabilitation training in future studies, especially for patients with brain injuries.
Point 4: The authors mention that parameters for RAGT were recorded, but they do not provide any information on what these parameters were or how they were used to guide the training. This information would be valuable for researcher who may be interested in replicating the study.
Response 4: Thank you for the comment. We have added information about parameters and how to guide patients to treatment as below.
Line 90-92: Patients walked at a self-selected speed during the treatment sessions, though they were encouraged to keep the speed as fast as possible. Maximal assist power was adjusted to emulate normal gait patterns according to a therapist’s inspection.
Point 5: The author reported that only two mild adverse events were reported and explained in the manuscript that the event caused mild pain in the patient during the experiment, but there were no details on how to handle it. This information is important for understanding the safety of wearable exoskeletons and should be included in the manuscript.
Response 5: Thank you for the comment. We have included information about patient’s management and condition as below.
Line 151-153: The pain was reduced by adjusting the position of the strap. Patients did not take any pain-relief medication due to the diminishing intensity of pain after cessation of the treatment session.
6.The sample size for this study was relatively small, including only 29 patients. The authors acknowledge this limitation, but lack a discussion on how it affects the extensibility of their findings.
Response 6: Thank you for the comment. As you have pointed out, small heterogenous sample size is limitation of our study. We have added some discussions as below.
Line 191-192: Our study should be interpreted with caution because it has a small sample size and is not a randomized controlled trial.
Line 194-197: Thus, prospective randomized controlled studies with larger numbers of patients are needed to confirm the effectiveness and safety of RAGT in patients with various clinical conditions.
Reviewer 2 Report
Dear authors.
Thank you for submitting your study
There are some mistakes in the number of patients. In line 14-17 and 98-99 the number or patients distributed by diagnoses are a total of 28. The same problem appears in the distribution by sex in line 98.
In table 2, BBS appears with ** (BBS**) but there is no explanation about the meaning of this mark.
Thank you very much
Author Response
Response to Reviewer 2 Comments
Point 1: There are some mistakes in the number of patients. In line 14-17 and 98-99 the number or patients distributed by diagnoses are a total of 28. The same problem appears in the distribution by sex in line 98.
Response 1: Thank you for the comment. We removed the mistakes as below.
line 14-17: Twenty-eight patients receiving more than ten sessions of overground RAGT using joint-torque-assist wearable exoskeletal robot were retrospectively analyzed in this study.
line 62-63: Twenty-eight patients were selected for the analysis based on the following inclusion criteria:
line 98-99: Twenty-eight patients were selected for the study with a mean (± SD) age of 43.9 ± 22.4 years (range, 19–86 years) and included 17 male and 11 female patients.
Point 2: In table 2, BBS appears with ** (BBS**) but there is no explanation about the meaning of this mark.
Response 2: Thank you for the comment. We added the explanation in Table 2.
Round 2
Reviewer 1 Report
I think the authors have revised the manuscript according to the review comments and agreed to accept this paper.